# Conductometric ppb-Level CO Sensors Based on In_2_O_3_ Nanofibers Co-Modified with Au and Pd Species

**DOI:** 10.3390/nano12193267

**Published:** 2022-09-20

**Authors:** Wenjiang Han, Jiaqi Yang, Bin Jiang, Xi Wang, Chong Wang, Lanlan Guo, Yanfeng Sun, Fangmeng Liu, Peng Sun, Geyu Lu

**Affiliations:** 1State Key Laboratory of Integrated Optoelectronics, Jilin Key Laboratory of Gas Sensors, College of Electronic Science and Engineering, Jilin University, 2699 Qianjin Street, Changchun 130012, China; 2College of Communication Engineering, Jilin University, Changchun 130022, China; 3School of Physics and Electronic Information Engineering, Henan Polytechnic University, Jiaozuo 454003, China; 4International Center of Future Science, Jilin University, Changchun 130012, China

**Keywords:** In_2_O_3_ nanofibers, carbon monoxide gas sensor, electrospinning, Au and Pd species co-modifying, low detection limit

## Abstract

Carbon monoxide (CO) is one of the most toxic gases to human life. Therefore, the effective monitoring of it down to ppb level is of great significance. Herein, a series of In_2_O_3_ nanofibers modified with Au or Pd species or simultaneous Au and Pd species have been prepared by electrospinning combined with a calcination process. The as-obtained samples are applied for the detection of CO. Gas-sensing investigations indicate that 2 at% Au and 2 at% Pd-co-modified In_2_O_3_ nanofibers exhibit the highest response (21.7) to 100 ppm CO at 180 °C, and the response value is ~8.5 times higher than that of pure In_2_O_3_ nanofibers. More importantly, the detection limit to CO is about 200 ppb with a response value of 1.23, and is obviously lower than that (6 ppm) of pure In_2_O_3_ nanofibers. In addition, the sensor also shows good stability within 19 days. These demonstrate that co-modifying In_2_O_3_ nanofibers with suitable amounts of Pd and Au species might be a meaningful strategy for the development of high-performance carbon monoxide gas sensors.

## 1. Introduction

The incomplete combustion of fuel in a closed environment, such as home stoves, engines, and fires, will inevitably produce CO, which has extremely higher affinity to human hemoglobin compared with O_2_, and threatens the safety of human life [1]. Currently, the number of poisoning incidents caused by CO is about 15,000 per year [2]. Therefore, it is urgent and meaningful to develop high-performance CO gas sensors with high response and low detection limit.

CO sensors based on metal oxide gas sensors have obvious cross-response to various reducing or combustible gases such as H_2_ and CH_4_, especially H_2_. During the use of fuel-cell vehicles using H_2_ as fuel, excessive CO will cause poisoning of the fuel cells and affect the operation of the vehicle [3]. In order to ensure the normal movement of fuel-cell vehicles, it is urgent to improve the selectivity of CO sensors to other combustible gases. Metal oxide semiconductors, such as SnO_2_ [4], In_2_O_3_ [5], and ZnO [6], have been mostly investigated for CO detection. Among them, In_2_O_3_ showed excellent selectivity for detecting CO compared with other oxides [7,8]. However, unmodified In_2_O_3_ has poor response to CO [9] and requires further modification to improve its sensing performance [5]. Au, Pd, and Pt [8] and their compounds are currently the most commonly used modified noble metals due to their excellent catalytic and electrical properties. Yin et al. synthesized SnO_2_ loaded with Au, Pd, and Pt by a sol-gel method. The results show that the loading of Au and Pd can enhance the selectivity of CO to H_2_, while the loading of Pt can enhance the selectivity of H_2_ to CO [8]. This indicates that the modification of Au and PdO can effectively improve the detection performance of gas sensors for CO, especially regarding selectivity. Hung et al. used Au-modified ZnO thin films to obtain a 6.4 response to 20 ppm CO at 250 °C [9]. Although the modification of Au can improve the response of the sensor to CO, the operating temperature of gas sensors is higher, to some degree. Wang et al. prepared PdO/SnO_2_ nanoparticles with optimal working temperature (OWT) at a low temperature of 100 °C, and the sensors exhibited good selectivity to CO [10]. This indicates that PdO modification can not only improve the selectivity of CO against H_2_, but can also effectively lower the OWT. In summary, Au might have the ability to adsorb more CO but with low catalytic effect, and a higher working temperature was necessary to obtain the maximal gas response. PdO species might have an excellent catalytic effect to CO [11,12], and can oxidize CO to CO_2_ at low temperatures or even room temperature [13]. Therefore, it is possible to fully utilize these two noble metals to obtain high-performance CO gas sensors with high response, good selectivity, and low OWT. However, only Pd–Au alloy has been used as modifying content for detecting CO, and the gas-sensing performance was not ideal [4]. As far as we know, an In_2_O_3_ gas sensor co-modified by Au and Pd has never been studied. Inspired by these ideas, we try to utilize the good selectivity of In_2_O_3_ combined with the co-modification of Au and PdO for the development of high-performance CO gas sensors.

Electrospinning technology has been widely used in the preparation of gas-sensing materials due to its unique one-dimensional structure with high electron mobility [14]. In this paper, In_2_O_3_ nanofibers were prepared by electrospinning and combined with the different properties of Au and PdO, two noble metals, and the modification effects of Au, PdO, and Au@PdO on CO were studied. Gas-sensing studies have shown that the co-modification of Au and PdO can not only greatly improve the gas response and selectivity to CO, but can also reduce the detection limit with a faster response speed. Such improvements in gas-sensing performance may be related to the combination of the adsorption ability of Au and the catalytic effect of PdO towards CO gas.

## 2. Experimental Section

### 2.1. Preparation of Pure In_2_O_3_, PdO-Modified In_2_O_3_, Au-Modified In_2_O_3_, and Au and PdO-Co-Modified In_2_O_3_ Nanofibers

All of the reagents were analytical grade and were used as received without further purification. In_2_O_3_ nanofibers with different Au and Pd modifying concentrations were synthesized by an electrospinning method, and then were calcinated in air to obtain the final products [15]. Firstly, solution A was prepared by dissolving 1 mmol In_2_O_3_∙4.5H_2_O and a certain dose of HAuCl_4_ and PdCl_2_ in a mixture solution containing 4 mL absolute ethanol and 4 mL N, N-Dimethylformamide (DMF), followed by continuous stirring at room temperature on a magnetic mixer for 20 min. The atomic ratios of ln, Au, and Pd in the six solutions A were 1:0:0:, 1:0.02:0, 1:0:0.02, 1:0.01:0.01, 1:0.02:0.02, and 1:0.04:0.04, respectively. After that, 1 g polyvinylpyrrolidone (PVP, Mw = 1,300,000 g/mol) was slowly added to solution A with a water bath for 5 h at 50 °C to obtain solution B. Subsequently, solution B was poured into a 5 mL plastic syringe for electrostatic spinning. The principles and use of typical electrostatic spinning equipment has been described in our previous work [15]. The related parameters were as follows: an electrostatic pressure of 17 kV was applied between the metal needle and collector with a distance of 15 cm, and the feed rate was controlled at 0.02 mL/h. The white as-spun precursor on aluminum foil was collected after 2 h, followed by calcination in a muffle furnace at 450 °C for 2 h with a heating rate of 2 °C/min. The samples were labeled as pure In_2_O_3_, Au2-In_2_O_3_, Pd2-In_2_O_3_, Au1Pd1-In_2_O_3_, Au2Pd2-In_2_O_3_, and Au4Pd4-In_2_O_3_ nanofibers.

### 2.2. Characterization

The X-ray powder diffraction (XRD) patterns of six groups of samples were analyzed using a X-ray diffractometer (XRD, Smartlab, Rigaku Corporation, Tokyo, Japan) in the 2θ range of 25°–80°. The morphology and size of the six groups of samples were observed by field emission scanning electron microscopy (SEM, Merlin Compact, Carl Zeiss NTS GmbH, Oberkochen, Germany) at an accelerating voltage of 15 kV. Further microscopic surface probing of pure In_2_O_3_ and Au2Pd2-In_2_O_3_ nanofibers was performed using a transmission electron microscope (TEM, Titan G260-300, FEI, Hillsboro, OR, USA) facility at an accelerating voltage of 200 kV. The sample binding energies and O 1s were analyzed by X-ray photoelectron spectroscopy (XPS, ESCALAB 250Xi, Thermo Scientifific K-Alpha, Waltham, MA, USA) using a Mg Kα (1358.6 eV) X-ray source.

### 2.3. Fabrication and Measurement of Gas Sensor

The traditional hexapod ceramic tube was used to make the sensor device. The actual image of the hexapod ceramic tube gas sensor and the illustration of the ceramic tube gold electrode are shown in Figure 1. The fabrication and testing process of the device has been introduced in detail in the previous study [16]. The sensor was aged for 48 h at an aging current of 100 mA under a laboratory environment (~25 °C, ~35% RH). The sensor was placed in a gas cylinder filled with air and the gas to be measured, and a Fluke (8846A) meter was used to detect the change in the resistance value at both ends of the Au electrode. The humidity test was carried out in a humidity chamber (Shanghai ESPC Environmental Equipment Company, Shanghai, China), and the temperature of the humidity chamber was always controlled at 25 °C. The gas response (S) is defined as the ratio of the stable resistance (R_a_) in air to the stable resistance (R_g_) in the gas being measured (S = R_a_/R_g_). Furthermore, response and recovery times are defined as the time required for a 90% change in the sensor resistance.

## 3. Results and Discussion

### 3.1. Structural and Morphological Characteristics

XRD patterns of all samples are presented in Figure 2a. According to XRD patterns (Figure 2a), all diffraction peaks of the as-obtained samples are consistent with the standard card of In_2_O_3_ (PDF#06-0416). The diffraction peaks belonging to Au and Pd species can be barely observed due to the low modifying contents of Au and Pd species. Figure 2b displays the patterns of the high resolution (440) peak of all samples. As shown in Figure 2b, the half widths of the diffraction peaks of the other samples gradually increase and shift to small angles with an increase in the modifying amount of Pd species [17]. This is because the radius of a Au atom (0.134 nm) is too large (larger than that of In^3+^(0.08 nm)) to incorporate into the In_2_O_3_ lattice, while the radius of Pd^2^^+^(0.085 nm) is only slightly larger than that of In^3+^, and Pd^2^^+^ can replace the position of In ions, which makes the interplanar spacing larger. The grain sizes are calculated according to Scherrer’s formula (D = 0.89 λ/β cosθ) (Table 1). It can be seen that Au modification can only slightly decrease the crystal size, while Pd^2+^ doping can significantly decrease it, indicating that Pd^2^^+^ might enter the In_2_O_3_ crystal and effectively inhibit the growth of In_2_O_3_ grain size. The reduced grain size is beneficial to the gas-sensing performance [16].

SEM images of the as-obtained samples demonstrate that the solo Au modification can cause the nanofibers to be obviously rougher (Figure 3c,d), while PdO modification cannot change the surface morphology significantly (Figure 3e,f) compared with pure In_2_O_3_ nanofibers (Figure 3a,b). The surface will become smoother with an increase in co-modifying amounts of Au and PdO (Figure 3i–l), which is due to a decrease in the grain size with the increase in the modifying amount. However, the nanofiber structure can be well-preserved after noble metal modification, and the target gas can easily transport in and out of the sensing body, which can ensure the high utility ratio of the gas sensors.

TEM images of pure In_2_O_3_ (Figure 4a) and Au2Pd2-In_2_O_3_ samples (Figure 4c) demonstrate that the surface of the nanofiber will become rougher after the addition of Au and Pd salts, which is in accordance with the SEM results (Figure 3b–j). The SAED image of the Au2Pd2-In_2_O_3_ sample is shown in Figure 4b. The (211), (222), (521), (541), and (655) lattice planes of In_2_O_3_, the (200) plane of Au, and (002), (110), and (212) planes of PdO lattice planes can be identified, which confirms the co-existence of Au, PdO, and In_2_O_3_ in the sensing material. The average lattice spacing of 0.416 nm, 0.200 nm, and 0.300 nm—which are in accordance with the (211) plane of the In_2_O_3_, (200) planes of Au, and (100) planes of PdO—can be clearly observed in Figure 4d,e. The results agree well with the SAED characterization, further confirming the co-existence of Au, PdO, and In_2_O_3_. Furthermore, elemental mapping images of the Au2Pd2-In_2_O_3_ sample demonstrate the uniform distribution of In, O, Au, and Pd elements as shown in Figure 4f–j.

XPS is carried out to analyze the surface chemical compositions as shown in Figure 5. The In 3d_5/2_ and In 3d_3/2_ peaks located at 444.13 eV and 451.68 eV for the pure In_2_O_3_ nanofiber shift to higher binding energies of 444.34 eV and 451.89 eV after noble metal modification as shown in Figure 5b, while the Au 4f_7/2_ and Au 4f_5/2_ peaks of the Au2Pd2-In_2_O_3_ sample located at 83.18 eV and 86.88 eV become lower compared with those of metallic Au located at 84.0 eV and 87.6 eV caused by the electron transfer from In_2_O_3_ to Au [18,19]. Figure 5d,e are the high-resolution Pd 3d spectra of the Pd2-In_2_O_3_ and Au2Pd2-In_2_O_3_ samples, indicating the successful Pd modification in In_2_O_3_ crystal [20]. Pd 3d_3/2_ can be divided into 341.10 eV (Pd^0^) and 342.40 eV (Pd^2+^), and Pd 3d_5/2_ can be decomposed into 335.08 eV (Pd^0^) and 336.90 eV (Pd^2+^). It is obvious that Pd species mainly existed in the form of Pd^2+^ in the Pd2-In_2_O_3_ sample (96.3%) and the Au2Pd2-In_2_O_3_ sample (85.6%). Many works have reported the co-existence of Pd species [21,22], which could lead to a significant increase in gas response. Figure 5f–i are high-resolution O 1s spectra resolved into three peaks, including lattice oxygen (O_L_) at 529.50 ± 0.6 eV, defective oxygen (O_D_) at 530.55 ± 0.4 eV, and adsorbed oxygen (O_C_) at 531.65 ± 0.3 eV [23]. It is clear that the relative percentages of O_D_ and O_C_ are significantly increased after noble metal modification, and the Au2Pd2-In_2_O_3_ sample possesses the highest relative percentages of O_D_ and O_C_. Therefore, the highest response of the Au2Pd2-In_2_O_3_ sample can be expected [24]. The percentages of the three different O 1s compositions in the four samples are listed in Table 2.

### 3.2. Sensing Performance

The responses to 100 ppm CO of the gas sensor based on as-obtained samples are shown in Figure 6a. It is clear that both Au modification (Au2-In_2_O_3_) and PdO modification (Pd2-In_2_O_3_) can increase the gas response to 100 ppm CO compared with pure In_2_O_3_, and the sensitization effect of Au is more prominent. However, Au modification does not change the OWT (300 °C), while PdO does lower the OWT to 150 °C [10]. Co-modification of Au and PdO into In_2_O_3_ nanofibers can further increase the gas response, and Au2Pd2-In_2_O_3_ possesses the highest response (21.7) among these sensors. Higher (Au4Pd4-In_2_O_3_, 11.2) and lower (Au1Pd1-In_2_O_3_, 11.6) co-modifying contents will decrease the gas response. The OWT of the gas sensors after co-modification of Au and PdO is 180 °C, located between 150 °C (Pd2-In_2_O_3_) and 300 °C (Au2-In_2_O_3_), indicating the synergistic effect of Au and PdO. The baseline resistance of the gas sensors based on the as-obtained samples will decrease with an increase in the working temperature (Figure 6b), which is a typical property of semiconductors.

Figure 7a–c show the dynamic response curves of pure In_2_O_3_ and Au2Pd2-In_2_O_3_ sensors at their optimal operating temperatures. The responses will increase with an increase in CO gas concentration for both sensors, and the resistances can recover their original value after exposure to different CO concentrations. The response values of the gas sensor based on pure In_2_O_3_ nanofibers to 6, 10, 20, 50, 100, 300, and 500 ppm CO gas at 300 °C are 1.19, 1.37, 1.56, 2.05, 2.30, 3.86, and 4.55, respectively. The response values based on the Au2Pd2-In_2_O_3_ sample to 0.2, 0.5, 1, 3, 6, 10, 20, 50, 100, 300, and 500 ppm CO gas at 180 °C are 1.23, 1.37, 1.40, 1.83, 2.42, 3.30, 5.44, 11.7, 21.7, 54.0, and 65.2, respectively. It is clear that the Au2Pd2-In_2_O_3_ sensor exhibits a higher gas response to the different concentrations of CO than the pure In_2_O_3_ sensor. In addition, the Au2Pd2-In_2_O_3_ sensor exhibits an extremely lower detection limit (0.2 ppm) than the pure In_2_O_3_ sensor (6 ppm), indicating that noble metal modification can not only enhance the gas response, but can also decrease the detection limit. Figure 7b and Figure 6d demonstrate that the gas responses share an almost linear relationship with CO concentration, which is beneficial for accurate gas concentration detection.

The selectivity of the gas sensors is shown in Figure 8. The ratio of the response values for CO and H_2_ is taken to evaluate the selectivity. It is clear that Au modification is more effective for increasing the CO gas response than PdO modification, but the selectivity of the Au-modified sensor (~1.3) is smaller than the PdO-modified sensor (~1.6). In addition, the response to benzene and toluene has also been significantly increased. Co-modification using Au and PdO for the Au1Pd1-In_2_O_3_ sensor can further increase the gas response to CO and H_2_ with higher selectivity (2.4), and suppress the response to benzene and toluene at the same time. The response for CO and H_2_ is further enhanced with an increase in Au and PdO modifying content (Au2Pd2-In_2_O_3_), and the selectivity is changed slightly (~2.2). At the same time, the response to benzene and toluene has been further suppressed. The response becomes worse and the selectivity becomes lower (1.5) when the co-modifying content of Au and PdO is too high (Au4Pd4-In_2_O_3_). Therefore, the Au2Pd2-In_2_O_3_ sensor is more suitable for CO detection in this work.

The response/recovery times for pure In_2_O_3_ (38 s/220 s) and Au2Pd2-In_2_O_3_ (5 s/106 s) at 180 °C can be observed in Figure 9a,b, indicating that noble metal modification can accelerate the response/recovery speed. Such an improvement in the response/recovery speed can be attributed to the catalytic effect of noble metal modification, which has been verified in many works [25,26]. The six reversible cycling dynamic response curves of Au2Pd2-In_2_O_3_ 100 ppm CO demonstrate that the Au2Pd2-In_2_O_3_ sensor has good reproducibility and repeatability (Figure 9c). The stability test of Au2Pd2-In_2_O_3_ for 19 days indicates that the baseline resistance in air and response to 100 ppm CO are almost unchanged (Figure 9d), which is of great significance in practical applications. Overall, the theAu2Pd2-In_2_O_3_ sensor has favorable sensing characteristics in its high gas response and low detection limit for CO detection compared with the existing reports as shown in Table 3.

Humidity is also an important factor affecting the sensing performance of oxide semiconductor sensors [30]. The dynamic response curves of the Au2Pd2-In_2_O_3_ sensor at 30%, 50%, 70%, and 90% RH were measured under 25 °C (Figure 10a), as well as the baseline resistance and response of the sensor versus humidity change (Figure 10b). When humidity was increased from 35% to 98% RH, baseline resistance of the sensor decreased by 42% and the response value decreased by 28%. This reduction in baseline resistance is due to the reaction of water molecules with oxygen on the surface of the material. Secondly, water molecules can compete with gas molecules and oxygen for adsorption sites on the surface of the sensing material, leading to a decrease in the response value [31]. Nevertheless, response of the Au2Pd2-In_2_O_3_ sensor to 100 ppm CO is still up to 15.2 under 98% RH, indicating excellent moisture resistance.

### 3.3. Sensing Mechanism

The gas-sensing mechanism can be explained by the space-charge layer, oxygen adsorption, and surface reaction between the reducing gases and adsorbed oxygen [32,33]. The sensing enhancement of the gas response can be mainly attributed to the electronic sensitization and chemical sensitization of the noble metals [24,34,35], as well as the synergistic effect of Au and PdO.

The electronic sensitization effect can be seen from the baseline resistance increase after noble metal modification, as shown in Figure 6b. The work functions of In_2_O_3_, Au, and PdO are about 4.3 eV [36], 5.1 eV [37], and 7.9 eV [38], respectively. The Schottky contact between Au and In_2_O_3_ and p–n heterojunction between PdO and In_2_O_3_ will formed after co-modification using Au and PdO [39,40,41]. The electrons will flow from In_2_O_3_ to Au and PdO and lead to the formation of the thicker space-charge layer due to the different Fermi levels of these materials as shown in Figure 11. Therefore, the baseline resistance will increase. In addition, Au modification seems to have better effect than PdO modification on the increase in the baseline resistance according to Figure 6b. This may be due to the fact that Au atoms mainly exist on the surface of In_2_O_3_ nanofibers and many Au-In_2_O_3_ contacts can be formed, while the majority of Pd ions enter into the In_2_O_3_ crystal and the formations of the p–n heterojunctions are fewer. However, the increase in baseline resistance after modification with Au and PdO is beneficial in improving the gas response [42].

The chemical sensitization can be verified by the XPS results as shown in Table 2. It is clear that relative chemisorbed oxygen (O_C_) percentages will significantly increase after noble metal modification, and Au2Pd2-In_2_O_3_ has the highest O_C_ content. Therefore, it is reasonable that the Au2Pd2-In_2_O_3_ sensor has the highest gas response. The OWT changes from 300 °C to 180 °C after co-modification using Au and PdO. The lower OWT also indicates the chemical sensitization effect, which leads to a higher gas response.

The synergistic effect of Au and PdO can be seen from the OWT after co-modification using Au and PdO. Solo Au modification will not change the OWT obviously (300 °C for Au2-In_2_O_3_), while solo PdO modification (Pd2-In_2_O_3_) will lower the OWT from 300 °C to 150 °C (Figure 6a). This indicates that the catalytic effect of PdO is much better than that of Au. The OWT after co-modification using Au and PdO is 180 °C. The intermediate OWT between 300 °C and 150 °C indicates the synergistic effect of Au and PdO. Furthermore, we speculate that Au might have higher adsorption ability than PdO to CO gas [43], which might be verified by the higher Oc content (35.2%) of Au2-In_2_O_3_ than that (33.6%) of Pd2-In_2_O_3_ as shown in Table 2, indicating that Au loading might adsorb more CO gas than Pd modification. However, a high OWT (300 °C) is necessary for solo Au modification to ensure the full reaction between CO gas and adsorbed oxygen due to the low catalytic effect of Au. After PdO modification, the gas response can be significantly increased though fully utilizing the adsorption ability of Au and catalytic effect of PdO with relatively lower OWT. Therefore, both the selectivity and gas response to CO gas can be greatly imporved. In addition, the synergistic effect can be also illustrated by the peak shift of Au 4f_7/2_ and Au 4f_5/2_ from 83.43 eV and 87.08 eV (Au2-In_2_O_3_, not shown here) to 83.18 eV and 86.88 eV (Au2Pd2-In_2_O_3_) as shown in Figure 5d, indicating greater electron transportation to the Au cluster after PdO doping. It is obvious that co-modification using suitable amounts of Au and PdO (Au2Pd2-In_2_O_3_) optimizes the electronic sensitization and chemical sensitization, reduces the activation energy of the reaction, greatly improves the detection limit of CO, and enhances the sensitivity of the material to CO gas.

## 4. Conclusions

In this work, the co-modifying effects of Au and PdO on In_2_O_3_ nanofibers are systematically investigated for the enhancement of CO detection compared with pure In_2_O_3_ and individual Au and PdO doping. The experimental results indicate that a suitable modifying amount of Au and PdO (Au2Pd2-In_2_O_3_) can not only increase the gas response, but can also lower the detection limit down to 200 ppb with a response value of 1.23 and OWT of 180 °C. The enhancement of the gas response to CO can be attributed to the electronic sensitization due to the formation of p–n heterojunctions between PdO and In_2_O_3_, Schottky contact between Au and In_2_O_3_, the chemical sensitization of PdO verified by lower OWT and higher Oc contents, and the synergistic effect of Au and PdO caused by the high adsorption ability of Au and excellent catalytic effect of PdO. The response to 100 ppm CO for the Au2Pd2-In_2_O_3_ sensor is about 8.5 times higher than that for the pure In_2_O_3_ sensor. The synergistic effect of Au and PdO also improves the selectivity to CO in different gases, especially the selectivity of CO against H_2_. Co-modification using Au and PdO might be a promising strategy for the enhancement of gas-sensing performance for CO detection.

## Figures and Tables

**Figure 1 nanomaterials-12-03267-f001:**
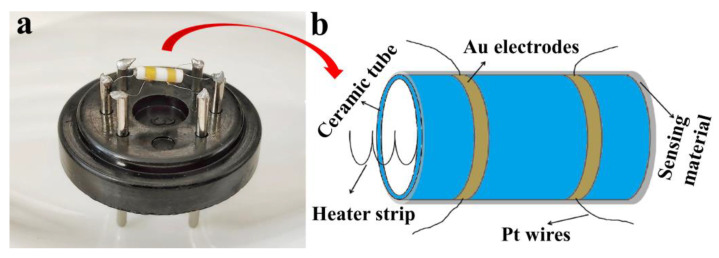
(**a**) Actual image of a hexapod ceramic tube gas sensor; (**b**) Illustration of ceramic tube Au electrode.

**Figure 2 nanomaterials-12-03267-f002:**
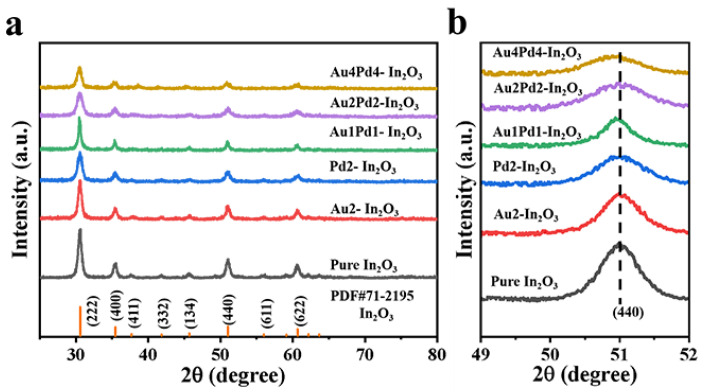
(**a**) XRD patterns and (**b**) high resolution of (440) peak of all the obtained samples.

**Figure 3 nanomaterials-12-03267-f003:**
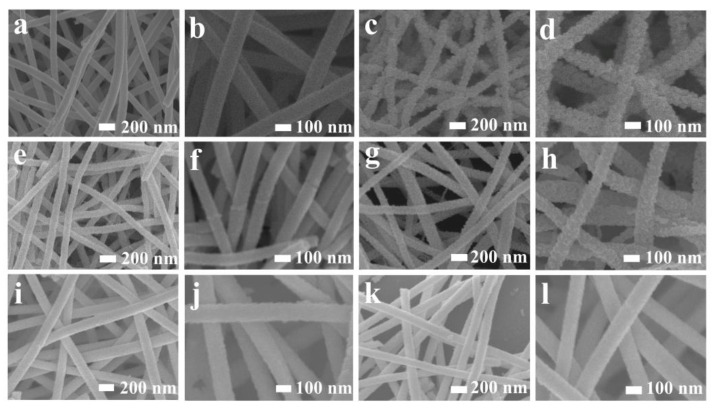
SEM images of (**a**,**b**) pure In_2_O_3_; (**c**,**d**) Au2-In_2_O_3_; (**e**,**f**) Pd2-In_2_O_3_; (**g**,**h**) Au1Pd1-In_2_O_3_; (**i**,**j**) Au2Pd2-In_2_O_3_; (**k**,**l**) Au4Pd4-In_2_O_3_ nanofibers.

**Figure 4 nanomaterials-12-03267-f004:**
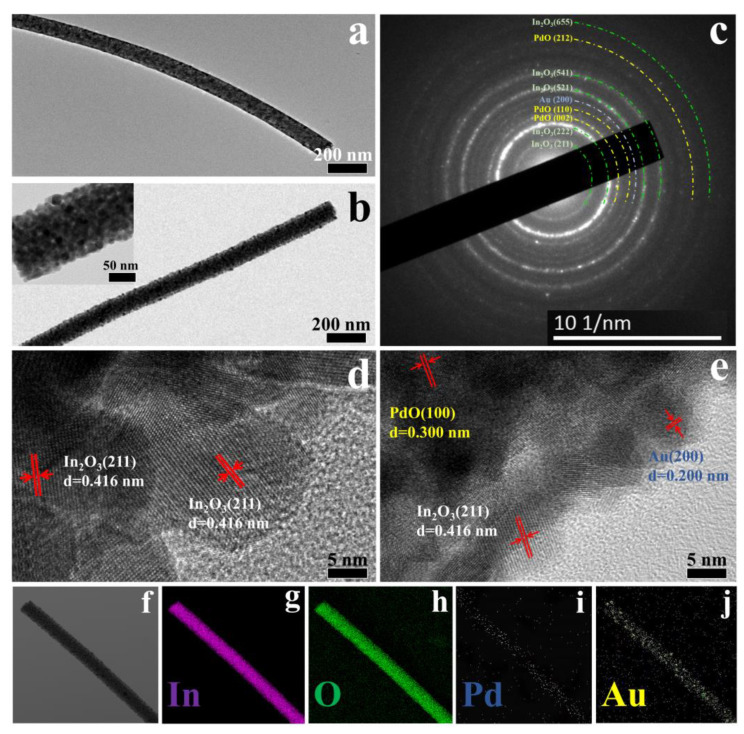
(**a**,**b**) TEM of pure In_2_O_3_ and Au2Pd2-In_2_O_3_; (**c**) SAED pattern of Au2Pd2-In_2_O_3_; (**d**,**e**) HRTEM of pure In_2_O_3_ and Au2Pd2-In_2_O_3_; (**f**–**j**) elemental mapping images of Au2Pd2-In_2_O_3_.

**Figure 5 nanomaterials-12-03267-f005:**
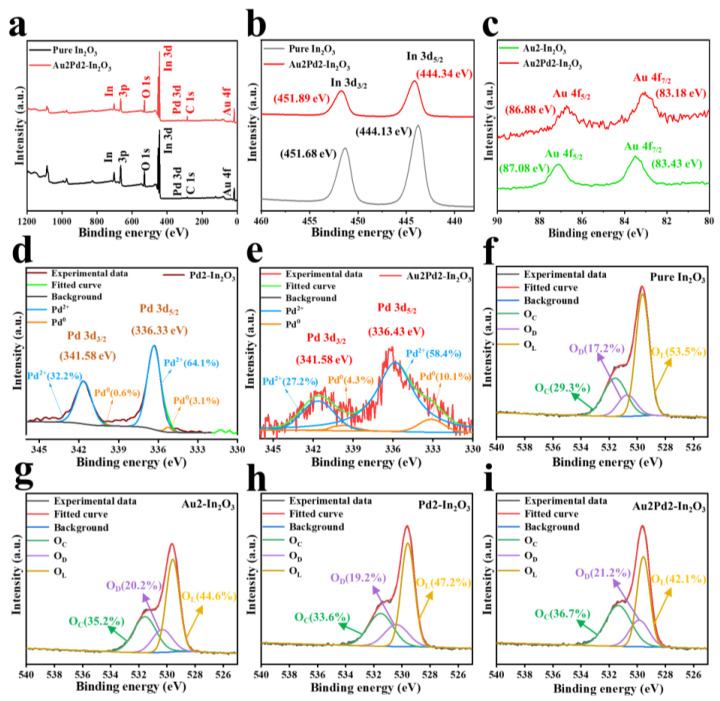
(**a**) XPS survey scan; high-resolution XPS spectra of (**b**) In 3d of pure and Au2Pd2-In_2_O_3_; (**c**) Au 4f Pd 3d of (**d**) pure and (**e**) Au2Pd2-In_2_O_3_; (**f**–**i**) O 1s XPS spectra of pure, Au2-In_2_O_3_, Pd2-In_2_O_3_, Au2Pd2-In_2_O_3_.

**Figure 6 nanomaterials-12-03267-f006:**
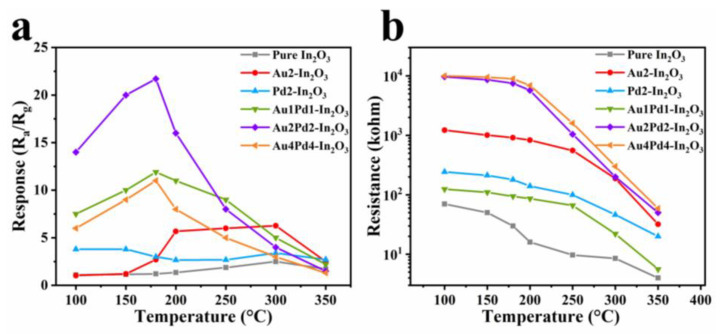
(**a**) Response values to 100 ppm CO gas; (**b**) baseline resistance values in air of nanofiber gas sensors based on six samples at different operating temperatures.

**Figure 7 nanomaterials-12-03267-f007:**
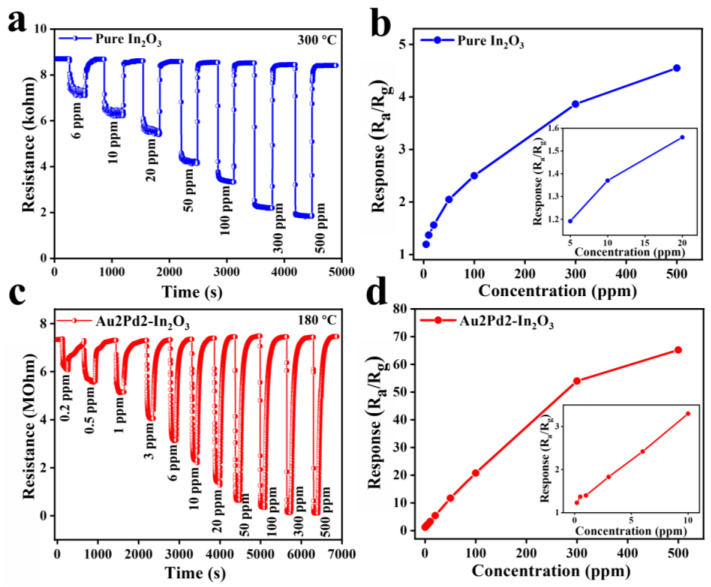
The sensor dynamic response curves of (**a**) pure In_2_O_3_ and (**c**) Au2Pd2-In_2_O_3_ samples; the corresponding response values of (**b**) pure In_2_O_3_ and (**d**) Au2Pd2-In_2_O_3_ samples at different concentrations of CO at 180 °C.

**Figure 8 nanomaterials-12-03267-f008:**
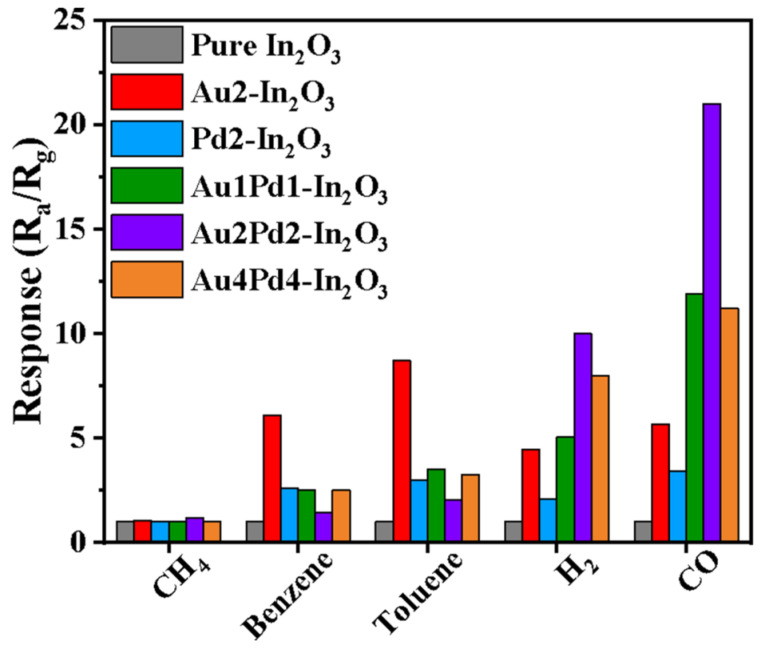
Gas responses of gas sensors based on six samples to various 100 ppm gases at 180 °C.

**Figure 9 nanomaterials-12-03267-f009:**
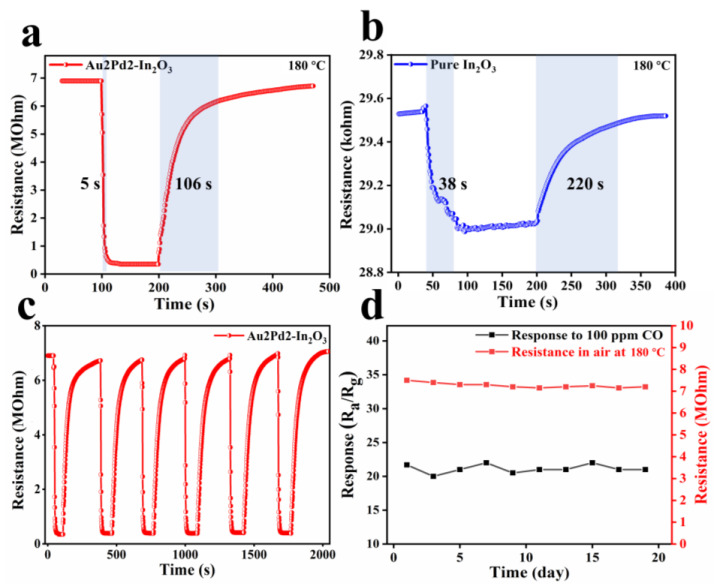
Dynamic response transients of sensors based on (**a**) Au2Pd2-In_2_O_3_ nanofibers and (**b**) pure In_2_O_3_ nanofibers to 100 ppm CO at 180 °C; (**c**) six dynamic cycling response curves and (**d**) one-week response and baseline resistance values of Au2Pd2-In_2_O_3_ sensor at 180 °C to 100 ppm CO.

**Figure 10 nanomaterials-12-03267-f010:**
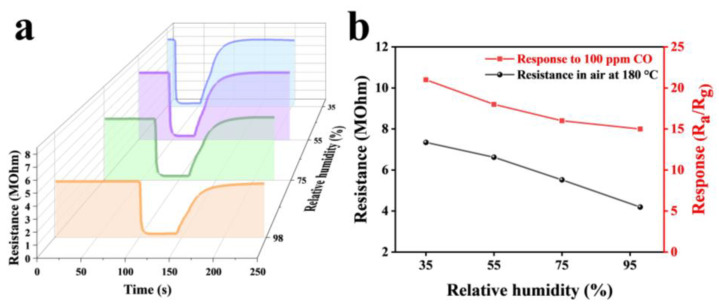
(**a**) The dynamic response curves and (**b**) the corresponding response and resistance relations of the sensors based on Au2Pd2-In_2_O_3_ nanofibers under different humidity conditions (35%, 55%, 75%, 98%) at the operating temperature of 200 °C.

**Figure 11 nanomaterials-12-03267-f011:**
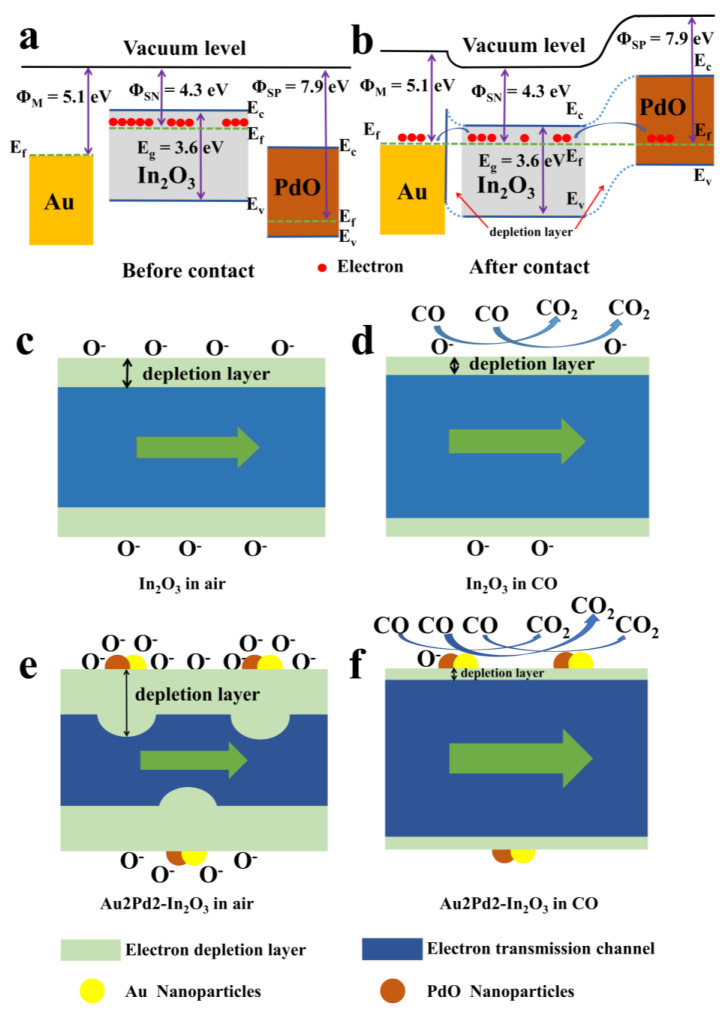
(**a**,**b**) energy-band diagrams of In_2_O_3_ before and after contact with Au and PdO; the schematic illustration of the sensing mechanism for (**c**,**d**) pure In_2_O_3_ and (**e**,**f**) Au2Pd2-In_2_O_3_ nanofibers to CO.

**Table 1 nanomaterials-12-03267-t001:** Grain size of all samples.

Sample	In_2_O_3_	Au2-In_2_O_3_	Pd2-In_2_O_3_	Au1Pd1-In_2_O_3_	Au2Pd2-In_2_O_3_	Au4Pd4-In_2_O_3_
grain sizes (nm)	16.2	15.9	12.0	13.8	10.4	10.2

**Table 2 nanomaterials-12-03267-t002:** Percentages of three different O 1s components in the four samples.

Sample	Lattice Oxygen O_L_ (%)	Defect Oxygen O_D_ (%)	Chemisorbed Oxygen O_C_ (%)	Sum of O_D_ and O_C_ (%)
Pure In_2_O_3_	53.5	17.2	29.3	46.5
Au2-In_2_O_3_	44.6	20.2	35.2	55.4
Pd2-In_2_O_3_	47.2	19.2	33.6	52.8
Au2Pd2-In_2_O_3_	42.1	21.2	36.7	57.9

**Table 3 nanomaterials-12-03267-t003:** Comparison of different gas sensors for detecting CO.

Materials	Morphology	T/°C and RH (%)	Conc. (ppm)	Response	LOD (ppm)	Res./Rec. Time (s)	Ref.
AuPd/SnO_2_	Nanoparticles	300 and 50	50	6 ^a^	50	-/-	[4]
Au/SnO_2_/In_2_O_3_	Nanofibers	200 and -	50	29 ^b^	10	80/400	[5]
Au/ZnO	Nanostars	35 and dry	50	15 ^a^	50	8/15	[6]
Pd/In_2_O_3_	Nanobundles	RT	100	12 ^c^	50	50/60	[27]
Pd/Fe/SnO_2_	Nanoparticles	350 and -	2000	22 ^a^	200	-/50	[28]
Pt/Co_3_O_4_/In_2_O_3_	Nano-branches	RT	100	~14 ^c^	5	~120/~20	[29]
**Au/PdO/In_2_O_3_**	**Nanofibers**	**180 and 30**	**100**	**21.7 ^a^**	**0.2**	**5/106**	**This work**

T and RH represents working temperature and humidity; conc. represents gas concentration; a: (R_a_/R_g_), b: ([R_a_−R_g_]/R_a_, %), c: (I_g_/I_a_); LOD represents low limit of detection; Res./Rec. time represents response/recovery time and Ref. represents reference.

## Data Availability

The data that support the findings of this study are available from the corresponding author upon reasonable request.

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
