# Peer review of "Conductometric ppb-Level CO Sensors Based on In2O3 Nanofibers Co-Modified with Au and Pd Species"

_nanomaterials, 2022, doi:10.3390/nano12193267_

Round 1

Reviewer 1 Report

Current manuscript focuses on the synthesis of Au and Pd co-doped In2O3 nanofibers for ehanced CO gas sensing properties. The approach by doping of noble metals into semiconductor metal oxides (SMOs) to enhance gas sensitivity is used widely in the field of SMO gas sensor. The results of manuscript shows that the response of sensors based on Pd co-doped In2O3 nanofibers is much higher than that of the sensors based on pristine In2O3 nanofibers and doping of individual elements into the In2O3 nanofibers. This is an interesting achievement. The manussript also provided detailed data and resuls on the characterization of materials and gas sensing properties. However, the manuscript still needs to improve several points to increase the quality of the manuscript.

-          The introduction part with the literature review is skethchy, thus, it is not clear about the scientific novelty of the manuscript.

-          Words “SEM imagines” may be not corrected for your mean in the line no,. 125 of the manuscript, please check carefully the manuscript again to avoid other typos

-          Based on HRTEM images, authors confirmed the existence of the Au and PdO in the In2O3 nanofibers and lattice spacing of PdO (100), Au (200); in fact, it not obviously to see the lattice spacing, please provide additional SAED images to confirm the statement.  

-          It is not clear to explain the faster response and recovery times of sensors based on the Au2Pd2-In2O3 nanofibers compared to the one based on In2O3 nanofibers, please discuss in more detail.

Author Response

Response to Reviewer Comments

Dear Reviewer:

  Thank you very much for give us an opportunity to resubmit our manuscript entitled “Conductometric ppb-level CO sensors based on In2O3 nanofibers with Au loading, Pd2+ doping and PdO loading”. We appreciate you very much for his positive and constructive comments and suggestions on our manuscript, and we have made the changes carefully according to your suggestions, which we wish to be considered for publication in “Nanomaterials”.

  We would like to resubmit the enclosed manuscript entitled “Conductometric ppb-level CO sensors based on In2O3 nanofibers with Au loading, Pd2+ doping and PdO loading”, hoping to get your consideration as much as possible to be accepted.

Thank you and best regards.

Sincerely yours

Yanfeng Sun and Wenjiang Han

Point 1: The introduction part with the literature review is sketchy, thus, it is not clear about the scientific novelty of the manuscript.

Response 1: Thank you for your meaningful suggestions. We provide detailed descriptions of the specific compounds such as SnO2 and ZnO doped with Au and Pd and Pt. The brief introductions about these works are also added in the revised manuscript. The influence of PdO and Au modification effect for CO detection has been discussed such as gas response and selectivity.

As far as we know, the gas sensors based on In2O3 co-modified by Au and PdO has never been studied. Inspired by these ideas, we try to utilize the good selectivity of In2O3 combined with the co-modifying of Au and PdO for the development of high-performance CO gas sensors. Such description has been added in the final part in the introduction section.

The modified parts are on lines 40-44, Line 46-51, line 60-67 and Line 68-76.

After revision:

Line 40-44

  Metal oxide semiconductors, such as SnO2 [4], In2O3 [5], ZnO [6] have been mostly investigated for CO detection. Among them, In2O3 showed excellent selectivity for detecting CO compared with other oxides [7,8]. However, the unmodified In2O3 has poor response to CO [9] and requires further modification to improve its sensing performance [5]. 

Line 46-51

  Yin et al. synthesized SnO2 loaded with Au, Pd, and Pt by a sol-gel method. The results show that the loading of Au, Pd can enhance the selectivity of CO to H2, while the loading of Pt can enhance the selectivity of H2 to CO [8]. This indicates that the modification of Au and PdO can effectively improve the detection performance of gas sensors for CO, especially the improvement of selectivity. Hung et al. used Au-modified ZnO thin films to obtain a 6.4 response to 20 ppm CO at 250°C [9].

line 60-67

  Therefore, it is possible to fully utilize these two noble metals to obtain high performance CO gas sensor with high response, good selectivity and low OWT. However, only PdAu alloy has been used as modifying content for detecting CO, and the gas sensing performance was not ideal [4]. As far as we know, the gas sensor of In2O3 co-modified by Au and Pd has never been studied. Inspired by these ideas, we try to utilize the good selectivity of In2O3 combined with the co-modifying of Au and PdO for the development of high-performance CO gas sensors.

Line 68-76

  Electrospinning technology has been widely used in the preparation of gas-sensing materials due to its unique one-dimensional structure with high electron mobility [14]. In this paper, In2O3 nanofibers were prepared by electrospinning and combined with the different properties of Au and PdO, two noble metals, and the modification effects of Au, PdO and Au@PdO on CO were studied. Gas sensing studies have shown that the co-modification of Au and PdO can not only greatly improve the gas response and selectivity to CO, but also reduce the detection limit with a faster response speed. Such improvement in gas sensing performance may be related to the combination of adsorption ability of Au and the catalytic effect of PdO towards CO gas.

Point 2: Words “SEM imagines” may be not corrected for your mean in the line no,. 125 of the manuscript, please check carefully the manuscript again to avoid other typos

Response 2: Thanks for your valuable comments. We are sorry for making a wrong description. We have corrected it to "SEM images". The full manuscript has also been checked carefully and some typos have been corrected.

After revision:

Line 143

  SEM images of the as-obtained sample demonstrate that the solo Au modifying can make the nanofibers obviously rougher (Fig. 3c-d), while PdO modifying can’t change the surface morphology significantly (Fig. 3e-f) compared with pure In2O3 nanofibers (Fig. 3a-b).

Point 3: Based on HRTEM images, authors confirmed the existence of the Au and PdO in the In2O3 nanofibers and lattice spacing of PdO (100), Au (200); in fact, it not obviously to see the lattice spacing, please provide additional SAED images to confirm the statement.

Response 3: Thank you for your professional suggestions about this work. We did the characterization of SAED, and the SAED images showed (002), (110) and (212) lattice planes of PdO, (211), (222), (521), (541) and (655) lattice planes of In2O3 and (200) lattice plane of Au. The SAED characterization image has also been added in Fig. 4 as Fig. 4c. Inaddition, the descriptions about this Fig. has been added in the manuscript. The modified parts are on lines 156-167 and in Fig. 4.

After revision:

   Line 156-167

  TEM images of pure In2O3 (Fig. 4a) and Au2Pd2-In2O3 samples (Fig. 4c) demonstrate that the surface of the nanofiber will become rougher after the addition of Au and Pd salts, which is in accordance with the SEM results (Fig. 3b-j). The SAED image of the Au2Pd2-In2O3 sample is shown in Fig. 4b. The (211), (222), (521), (541) and (655) lattice planes of In2O3, the (200) plane of Au and (002), (110) and (212) planes of PdO lattice planes can be identified, which confirm the co-exist of Au, PdO and In2O3 in the sensing material. The average lattice spacing of 0.416 nm, 0.200 nm and 0.300 nm, which are in accordance with the (211) plane of the In2O3, (200) planes of Au and (100) planes of PdO, can be clearly observed in Fig. 4d-e. The results agree well with the SAED characterization, further confirming the co-exist state of Au, PdO and In2O3.

Figure 4. (a,b) TEM of pure In2O3 and Au2Pd2-In2O3; (c) SAED pattern of Au2Pd2-In2O3, (d,e) HRTEM of pure In2O3 and Au2Pd2-In2O3; (f–j) elemental mapping images of Au2Pd2-In2O3.

Point 4: It is not clear to explain the faster response and recovery times of sensors based on the Au2Pd2-In2O3 nanofibers compared to the one based on In2O3 nanofibers, please discuss in more detail.

Response 4: Thank you for your valuable advice. We explain the acceleration in response recovery time. The response/recovery times of pure In2O3 and Au2Pd2-In2O3 have been compared at the same working temperature. It is clear that both the response and recovery speeds become faster after Au and PdO modification. The main reason for the faster speed can be mainly attributed to the catalytic effect of PdO. The reason has been added in the manuscript. Corresponding reference has also been added in the manuscript. The modified parts are on line 243-245.

After revision:

line 243-245

  Such improvement of response/recovery speed can be attributed to the catalytic effect noble metal modification, which has been verified in many works [23,24].

[23] S.W. Choi, A. Katoch, G.J. Sun, S.S. Kim, Bimetallic Pd/Pt nanoparticle-functionalized SnO2 nanowires for fast response and recovery to NO2, Sens. Actuators B Chem., 181(2013) 446-53.

[24] X. Wang, W.J. Han, J.Q. Yang, P.F. Cheng, Y.L. Wang, C.H. Feng, et al., Conductometric ppb-level triethylamine sensor based on macroporous WO3-W18O49 heterostructures functionalized with carbon layers and PdO nanoparticles, Sens. Actuators B Chem., 361(2022).

Thanks again for your hard work during the COVID pandemic.

Reviewer 2 Report

Authors are studied on the co-doped effect of Au and Pd on In2O3 nanofiber for conductometric based sensors for detection of CO gas. Although this study has been systematically conducted, some experimental data and technical explanations are still lacking. The following questions and comments should be revised carefully for publication in this high reputable journal.

1) Authors said in Fig2 that “the atomic size of Au atom (0.134 nm) is too larger than that of In3+(0.08 nm) to incorporate into the In2O3 lattice”. It this is the case then how it is possible to say Au and Pd co-doped In2O3 as mentioned in the title? I feel that the Au is just deposited on In2O3 rather than doped into the lattice of In2O3. Hence I strongly recommend to change the title and also clearly explain the XRD data accordingly. In addition, In HR-TEM also, PdO was detected rather than metallic Pd.

2) In XPS result, author also should clearly discuss about the Pd ion states in or on In2O3 nanofibers. If metallic Pd and PdO is coexisted in the In2O3 nanofibers, the gas sensing mechanism should be also changed.

3) The authors did not clearly explain the reason for the selectivity of Au and Pd co-doped In2O3 nanofiber sensor for CO detection. Please give scientific reason.

Author Response

Response to Reviewer Comments

Dear Reviewer:

  Thank you very much for give us an opportunity to resubmit our manuscript entitled “Conductometric ppb-level CO sensors based on In2O3 nanofibers with Au loading, Pd2+ doping and PdO loading”. We appreciate you very much for his positive and constructive comments and suggestions on our manuscript, and we have made the changes carefully according to your suggestions, which we wish to be considered for publication in “Nanomaterials”.

  We would like to resubmit the enclosed manuscript entitled “Conductometric ppb-level CO sensors based on In2O3 nanofibers with Au loading, Pd2+ doping and PdO loading”, hoping to get your consideration as much as possible to be accepted.

Thank you and best regards.

Sincerely yours

Yanfeng Sun and Wenjiang Han

Point 1: Authors said in Fig2 that “the atomic size of Au atom (0.134 nm) is too larger than that of In3+(0.08 nm) to incorporate into the In2O3 lattice”. It this is the case then how it is possible to say Au and Pd co-doped In2O3 as mentioned in the title? I feel that the Au is just deposited on In2O3 rather than doped into the lattice of In2O3. Hence I strongly recommend to change the title and also clearly explain the XRD data accordingly. In addition, In HR-TEM also, PdO was detected rather than metallic Pd.

Response 1: Thank you for your professional suggestions about this work. Au might be loaded on the surface of the sensing materials of In2O3. About Pd element, part of it will enter into the crystal lattice which will lead to the shift of the diffraction peaks to the low angle, and others will form the PdO materials deposited on the surface of the sensing material. Therefore, we have changed the title of the manuscript as “Conductometric ppb-level CO sensors based on In2O3 nanofibers with Au loading, Pd2+ doping and PdO loading”. The SAED characterization has been added in the manuscript. PdO, Au and In2O3 can be clearly observed without the detection of metal Pd. Corresponding descriptions has also been changed. The modified parts are on lines 127-139, 154-165.  

After revision (Modified parts are marked in red font):

XRD: lines 127-139

  The diffraction peaks belonging to Au and Pd species can be barely observed due to the low modifying contents of Au and Pd species. As shown in Fig. 2b, the half widths of the diffraction peaks of the other samples gradually increase and shift to small angles with the increase of the modifying amount of Pd species [17]. It is because that the radius of Au atom (0.134 nm) is too larger than that of In3+(0.08 nm) to incorporate into the In2O3 lattice, while the radius of Pd2+(0.085 nm) is only slightly larger than that of In3+, and Pd2+can replace the position of In ions, which makes the interplanar spacing larger. The grain sizes are calculated according to Scherrer's formula (D = 0.89 λ/β cosθ) (Table 1). It can be seen that Au modification can only slightly decrease the crystal size, while Pd2+ doping can significantly decrease it, indicating that Pd2+ might enter the In2O3 crystal and effectively inhibit the growth of In2O3 grain size. The reduced grain size is beneficial to the gas sensing performance [16].

TEM: lines 154-165

  TEM images of pure In2O3 (Fig. 4a) and Au2Pd2-In2O3 samples (Fig. 4c) demonstrate that the surface of the nanofiber will become rougher after the addition of Au and Pd salts, which is in accordance with the SEM results (Fig. 3b-j). The SAED image of the Au2Pd2-In2O3 sample is shown in Fig. 4b. The (211), (222), (521), (541) and (655) lattice planes of In2O3, the (200) plane of Au and (002), (110) and (212) planes of PdO lattice planes can be identified, which confirm the co-exist of Au, PdO and In2O3 in the sensing material. The average lattice spacing of 0.416 nm, 0.200 nm and 0.300 nm, which are in accordance with the (211) plane of the In2O3, (200) planes of Au and (100) planes of PdO, can be clearly observed in Fig. 4d-e. The results agree well with the SAED characterization, further confirming the co-exist state of Au, PdO and In2O3. Furthermore, elemental mapping images of the Au2Pd2-In2O3 sample demonstrate the uniform distribution of In, O, Au and Pd elements as shown in (Fig. 4f-j).

Point 2: In XPS result, author also should clearly discuss about the Pd ion states in or on In2O3 nanofibers. If metallic Pd and PdO is coexisted in the In2O3 nanofibers, the gas sensing mechanism should be also changed.

Response 2: Thanks for your valuable comments. About Pd element, part of it will enter into the crystal lattice which will lead to the shift of the diffraction peaks to the low angle as shown in the XRD characterization, and others will form the PdO materials deposited on the surface of the sensing material verified by the TEM characterization. However, metallic Pd can’t be observed in the above characterization. Even though the XPS result about Pd might be divided into the peak concerning metallic Pd, the portion of metallic Pd is very small and the fluctuation is large. Therefore, the effect of metallic Pd can be ignored in the manuscript. In the future works, we will pay more attention according to your meaningful suggestion.

Point 3: The authors did not clearly explain the reason for the selectivity of Au and Pd co-doped In2O3 nanofiber sensor for CO detection. Please give scientific reason.

Response 3: Thank you for your meaningful suggestions. Solo Au or PdO modification effect for the enhancement of CO selectivity has been added in the introduction part. Some works about this subject has also been added. In the sensing mechanism part, we think that the synergistic effect of Au and PdO modification might be the main reason for the enhancement for both selectivity and gas response to CO. therefore, the final part in the sensing mechanism has been further modified to provide some scientific reasons about the selectivity of Au and Pd modified In2O3 nanofiber sensor for CO detection. The modified parts are on lines 46-55 and 311-327.

After revision:

Line 46-55

  Yin et al. synthesized SnO2 loaded with Au, Pd, and Pt by a sol-gel method. The results show that the loading of Au, Pd can enhance the selectivity of CO to H2, while the loading of Pt can enhance the selectivity of H2 to CO [8]. This indicates that the modification of Au and PdO can effectively improve the detection performance of gas sensors for CO, especially the improvement of selectivity. Hung et al. used Au-modified ZnO thin films to obtain a 6.4 response to 20 ppm CO at 250°C [9]. Although the modification of Au can improve the response of the sensor to CO, the operating temperature of gas sensors is higher in some degree. Wang et al. prepared PdO/SnO2 nanoparticles with optimal working temperature (OWT) at a low temperature of 100°C, and the sensors exhibited good selectivity to CO [10].

Line 311-327

  Furthermore, we speculate that Au might have higher adsorption ability than PdO to CO gas [41], which might be verified by the higher Oc content (35.2%) of Au2-In2O3 than that (33.6%) of Pd2-In2O3 as shown in Table 2, indicating that Au loading might adsorb more CO gas than Pd modification. However, high OWT (300°C) is necessary for the sole Au modifying to ensure the full reaction between CO gas and adsorbed oxygen due to the low catalytic effect of Au. After PdO modifying, the gas response can be significantly increased though fully utilizing the adsorption ability of Au and catalytic effect PdO with relatively lower OWT. Therefore, both the selectivity and gas response to CO gas can be greatly imporved. In addition, the synergistic effect can be also illustrated by peak shift of Au 4f7/2 and Au 4f5/2 from 83.43 eV and 87.08 eV (Au2-In2O3, not shown here) to 83.18 eV and 86.88 eV (Au2Pd2-In2O3) as shown in Fig. 5d, indicating the more electron transportation to Au cluster after PdO doing. It is obvious that the co-modifying of suitable amount of Au and PdO (Au2Pd2-In2O3) optimizes the electronic sensitization and chemical sensitization, reduces the activation energy of the reaction, greatly improves the detection limit of CO, and enhances the sensitivity of the material to CO gas.

Reviewer 3 Report

The manuscript of Wenjiang Han et al, illustrates a series of In2O3 nanofibers doped with Au, Pd and Au/Pd as gas sensors for CO. The results are reasonable when compared with the existing data. In my opinion, the manuscript is suitable for publication after some modifications.

The introduction should have more details, especially lines 41-44, for instance, what type of compound is doped.

Also, the lines 34-37, “Metal oxide semiconductors, such as SnO2 [3], In2O3 [4], ZnO have been mostly investigated for CO detection. Among them, In2O3 seem to have the superior selectivity for CO detection [6,7]. Pure In2O3 [8] has poor response to CO and further modification is necessary to improve the sensing performance”. These paragraphs should be better explained, it is a little confusing.

 The Experimental Section “Firstly, solution A was prepared by dissolving 1 mmol In2O3∙4.5H2O and a certain dose of AuCl4H (0.02mmol), PdCl2 (0.02), 0.01 mmol AuCl4H and 0.01 mmol PdCl2, 0.02 mmol AuCl4H and 0.02 mmol PdCl2, and 0.04 mmol AuCl4H and 0.04 mmol PdCl2 in..”. It looks like all those reagents are added, for what I realized from the description it is not what happened. In my opinion, the author should separate each of the nanofibers.

No reference in the text is made to Figure 2b, it should be added.

Some formatting issues should be corrected, the legend of Figure 7 and the title of Table 3 are on another page.

Why do the authors choose for the selectivity studies the H2 and the other gases shown in Figure 8 ?

Author Response

Dear Reviewer:

     Thank you very much for give us an opportunity to resubmit our manuscript entitled “Conductometric ppb-level CO sensors based on In2O3 nanofibers with Au loading, Pd2+ doping and PdO loading”. We appreciate you very much for his positive and constructive comments and suggestions on our manuscript, and we have made the changes carefully according to your suggestions, which we wish to be considered for publication in “Nanomaterials”.

     We would like to resubmit the enclosed manuscript entitled “Conductometric ppb-level CO sensors based on In2O3 nanofibers with Au loading, Pd2+ doping and PdO loading”, hoping to get your consideration as much as possible to be accepted.

Thank you and best regards.

Sincerely yours

Yanfeng Sun and Wenjiang Han

Response to Reviewer  Comments

Point 1: The introduction should have more details, especially lines 41-44, for instance, what type of compound is doped.

Response 1: Thanks for your valuable comments. We provide detailed descriptions of the specific compounds such as SnO2 and ZnO doped with Au and Pd and Pt. The brief introductions about these works are also added in the revised manuscript. The modified part is in lines 46-55 respectively.

After revision (Modified parts are marked in red font):

Line 46-55

     Yin et al. synthesized SnO2 loaded with Au, Pd, and Pt by a sol-gel method. The results show that the loading of Au, Pd can enhance the selectivity of CO to H2, while the loading of Pt can enhance the selectivity of H2 to CO [8]. This indicates that the modification of Au and PdO can effectively improve the detection performance of gas sensors for CO, especially the improvement of selectivity. Hung et al. used Au-modified ZnO thin films to obtain a 6.4 response to 20 ppm CO at 250°C [9]. Although the modification of Au can improve the response of the sensor to CO, the operating temperature of gas sensors is higher in some degree. Wang et al. prepared PdO/SnO2 nanoparticles with optimal working temperature (OWT) at a low temperature of 100°C, and the sensors exhibited good selectivity to CO [10].

Point 2: Also, the lines 34-37, “Metal oxide semiconductors, such as SnO2 [3], In2O3 [4], ZnO have been mostly investigated for CO detection. Among them, In2O3 seem to have the superior selectivity for CO detection [6,7]. Pure In2O3 [8] has poor response to CO and further modification is necessary to improve the sensing performance”. These paragraphs should be better explained, it is a little confusing.

Response 2: Thanks for your professional comments. We have made an overall revision to the introduction section. The descriptions have been moved to the second paragraph. The selectivity of various metal oxide sensors to CO has been addressed according to the reported literature. The response values of these oxides have been added in the manuscript. Furthermore, the advantage of Au and Pd doping has been summarized. The modified parts are in lines 40-44, 46-51 and 57-62.  

After revision:

Line 40-44

     Metal oxide semiconductors, such as SnO2 [4], In2O3 [5, 9], ZnO [6] have been mostly investigated for CO detection. Among them, In2O3 showed excellent selectivity for detecting CO compared with other oxides [7,8]. However, the unmodified In2O3 has poor response to CO [9] and requires further modification to improve its sensing performance [5].

Line 46-51

     Yin et al. synthesized SnO2 loaded with Au, Pd, and Pt by a sol-gel method. The results show that the loading of Au, Pd can enhance the selectivity of CO to H2, while the loading of Pt can enhance the selectivity of H2 to CO [8]. This indicates that the modification of Au and PdO can effectively improve the detection performance of gas sensors for CO, especially the improvement of selectivity. Hung et al. used Au-modified ZnO thin films to obtain a 6.4 response to 20 ppm CO at 250°C [9].

Line 57-62

     In summary, Au might have the ability to adsorb more CO but with low catalytic effect, and higher working temperature was necessary to obtain the maximal gas response. PdO species might have the excellent catalytic effect to CO [11,12], which can oxidize CO to CO2 at low temperature or even room temperature [13]. Therefore, it is possible to fully utilize these two noble metals to obtain high performance CO gas sensor with high response, good selectivity and low OWT.

Point 3: The Experimental Section “Firstly, solution A was prepared by dissolving 1 mmol In2O3∙4.5H2O and a certain dose of AuCl4H (0.02mmol), PdCl2 (0.02), 0.01 mmol AuCl4H and 0.01 mmol PdCl2, 0.02 mmol AuCl4H and 0.02 mmol PdCl2, and 0.04 mmol AuCl4H and 0.04 mmol PdCl2 in..”. It looks like all those reagents are added, for what I realized from the description it is not what happened. In my opinion, the author should separate each of the nanofibers.

Response 3: Thank you for your valuable comments. We describe the experimental process separately in the revised manuscript. The revised sections are in lines 83-88.

After revision:

Line 83-88

     Firstly, solution A was prepared by dissolving 1 mmol In2O3∙4.5H2O and a certain dose of AuCl4H, PdCl2, in a mixture solution containing 4 mL absolute ethanol and 4 mL N, N-Dimethylformamide (DMF), followed by continuously stirring at room temperature on a magnetic mixer for 20 minutes. The atomic ratios of ln, Au and Pd in the six solutions A are 1:0:0:, 1:0.02:0, 1:0:0.02, 1:0.01:0.01, 1:0.02:0.02 and 1:0.04:0.04, respectively.

Point 4: No reference in the text is made to Figure 2b, it should be added.

Response 4: Thank you for your meaningful suggestions. We have added a reference to Figure 2b which shows the reason for the shift of the diffraction peaks. The modified parts are on lines 129-131.

After revision:

line 129-131

     As shown in Fig. 2b, the half widths of the diffraction peaks of the other samples gradually increase and shift to small angles with the increase of the modifying amount of Pd species [17].

[17] M. Yang, J.Y. Lu, X. Wang, H. Zhang, F. Chen, J.B. Sun, et al., Acetone sensors with high stability to humidity changes based on Ru-doped NiO flower-like microspheres, Sens. Actuators B Chem., 313(2020).

Point 5: Some formatting issues should be corrected, the legend of Figure 7 and the title of Table 3 are on another page.

Response 5: Thanks for your valuable comment. We have made changes to formatting issues such as the legend of Figure 7 and the title of Table 3.

Point 6: Why do the authors choose for the selectivity studies the H2 and the other gases shown in Figure 8 ?

Response 6: Thanks for your professional comments. About the fuel cell vehicles using H2 as fuel, CO gas will poison the fuel cells and effect the operation of the vehicle. Therefore, the development of the CO sensors with superior selectivity against H2 is necessary. Volatile organic compounds are also common interfering gases with generally higher response values than CO. Therefore, we have chosen H2 and some volatile organic compounds as the interfering gases in the manuscript. Especially, the necessity about the selectivity against H2 has been added in the introduction part. The modified parts are on lines 35-40.

After revision: 

line 35-40

     CO sensors based on metal oxide gas sensors have obvious cross-response to various reducing or combustible gases such as H2 and CH4, especially H2. During the use of fuel cell vehicles using H2 as fuel, excessive CO will cause poisoning of the fuel cells and affect the operation of the vehicle [3]. In order to ensure the normal movement of fuel cell vehicles, it is urgent to improve the selectivity of CO sensors to other combustible gases. 

Round 2

Reviewer 1 Report

Authors tried to improved the quality of the manuscript. I can can recommend to publish on Nanomaterials

Author Response

Thank you very much for your recommendation!

Reviewer 3 Report

The authors have made the suggested modifications. 

Author Response

Thank you very much for your valuable comments on our research.